# Easy Rocap: A Low-Cost and Easy-to-Use Motion Capture System for Drones

Haoyu Wang [1,2,3], Chi Chen [1,2,3,*], Yong He [1,2,3], Shangzhe Sun [1,2,3], Liuchun Li [4], Yuhang Xu [1,2,3] and Bisheng Yang [1,2,3]

1 State Key Laboratory of Information Engineering in Surveying, Mapping and Remote Sensing, Wuhan University, Wuhan 430072, China; spacewang@whu.edu.cn (H.W.); yong.he@whu.edu.cn (Y.H.); ssz@whu.edu.cn (S.S.); yuhangxu@whu.edu.cn (Y.X.); bshyang@whu.edu.cn (B.Y.)
2 Engineering Research Centre for Spatio-Temporal Data Acquisition and Smart Application (STSA), Ministry of Education in China, Wuhan 430072, China
3 Institute of Artificial Intelligence in Geomatics, Wuhan University, Wuhan 430072, China
4 Institute of Artificial Intelligence, School of Computer Science, Wuhan University, Wuhan 430072, China; liuc.lee@whu.edu.cn
* Correspondence: chichen@whu.edu.cn

**Abstract:** Fast and accurate pose estimation is essential for the local motion control of robots such as drones. At present, camera-based motion capture (Mocap) systems are mostly used by robots. However, this kind of Mocap system is easily affected by light noise and camera occlusion, and the cost of common commercial Mocap systems is high. To address these challenges, we propose Easy Rocap, a low-cost, open-source robot motion capture system, which can quickly and robustly capture the accurate position and orientation of the robot. Firstly, based on training a real-time object detector, an object-filtering algorithm using class and confidence is designed to eliminate false detections. Secondly, multiple-object tracking (MOT) is applied to maintain the continuity of the trajectories, and the epipolar constraint is applied to multi-view correspondences. Finally, the calibrated multi-view cameras are used to calculate the 3D coordinates of the markers and effectively estimate the 3D pose of the target robot. Our system takes in real-time multi-camera data streams, making it easy to integrate into the robot system. In the simulation scenario experiment, the average position estimation error of the method is less than 0.008 m, and the average orientation error is less than 0.65 degrees. In the real scenario experiment, we compared the localization results of our method with the advanced LiDAR-Inertial Simultaneous Localization and Mapping (SLAM) algorithm. According to the experimental results, SLAM generates drifts during turns, while our method can overcome the drifts and accumulated errors of SLAM, making the trajectory more stable and accurate. In addition, the pose estimation speed of our system can reach 30 Hz.

**Keywords:** motion capture; robot; UAV; UGV; LiDAR; SLAM; point cloud

## 1. Introduction

The rapid technological advances in robotic systems have inspired the development of unmanned platforms, including unmanned ground vehicles (UGVs), unmanned aerial vehicles (UAVs), and quadruped robots [1,2]. Pose estimation plays an important role in robotic applications. Without accurate estimation of position and velocity, a robot is unable to build the correct spatial representation of the environment, which leads to the impossibility of trajectory planning and execution [3]. In open outdoor environments, GPS and inertial measurement unit (IMU)-integrated navigation technology can provide pose information effectively [4,5]. However, in indoor environments, GPS signals attenuate severely, leading to inaccuracies in the provided location information [6,7]. Therefore, real-time pose information remains particularly crucial for most mobile robots.

There is a large amount of research exploring indoor positioning technology for robots, including visible light positioning (VLP), computer vision [8], ultrasonic [9], Bluetooth [10], and so on. Among them, both VLP and computer vision approaches reach high accuracy at a low cost [7]. Previous research has achieved speed improvements in VLP. However, this approach requires the unmanned device to carry heavy industrial cameras and high-performance processors [11], which is a challenge for payload-limited drones and other small robots [12,13]. Some computer-vision-based approaches utilize depth cameras to estimate the position changes of drones [14], but they are unsuitable for small-scale experiments with high precision requirements because of the limited positioning range and the significant dependency on the target detection result of drones [15]. Recently, with the rapid technological advances in computer hardware, the cost of visual positioning has been significantly reduced, and the rapid progress in visual algorithms has made real-time positioning possible. However, these positioning methods need to integrate other data like IMU data to obtain the position and orientation, which makes the approaches complex.

Simultaneous localization and mapping (SLAM) provides a framework for positioning and mapping, and a variety of technical methods have emerged in recent years [16,17]. Xu et al. [18] proposed a method of tightly coupling multiple LiDARs with IMU data for odometry, significantly improving the accuracy of pose estimation. Sun et al. [19] achieved self-pose estimation in weak GNSS environments by integrating GNSS, RGB camera, and IMU data on the drone. However, there are several challenges for SLAM on lightweight UAV platforms: due to the limited load, it is difficult to have sensors and computers of the same quality as those on ground robots [20]; UAVs have six degrees of freedom (DOF) poses, and cannot make some simplified assumptions like wheeled robots. At the same time, autonomous UAVs require high-frequency and real-time pose information to ensure stable control of the platform. Given these challenges, as well as the need for true pose data for indoor SLAM accuracy evaluation, it is essential to study a low-cost and high-precision method for indoor UAV pose estimation.

Motion capture (Mocap) systems based on cameras have better performance in robotic applications [21,22]. Most of these systems are commercial Mocap systems, such as Vicon, OptiTrack, and so on. However, commercial Mocap systems in the market often have a high cost and lack open-source software and hardware, making it difficult to deploy them on robots. The open-source Mocap toolbox Easy Mocap (https://github.com/zju3dv/EasyMocap (accessed on 29 March 2021)) provides the technology for capturing human body motions using multiple RGB cameras. However, mobile unmanned platforms such as UGV and UAV do not have structural key points with fixed connections like human joints. Therefore, markerless Mocap methods based on deep learning are inefficient, and it is difficult to directly apply Easy Mocap to robots. Utilizing infrared (IR) camera-based Mocap systems enables the position triangulation of markers on the target robot, which can be used for pose estimation of rigid bodies [23]. However, unlike the typical Mocap experimental environment with a single background and controlled lighting, the actual working environment of unmanned platforms is complex. IR Mocap systems are susceptible to environmental light noise, and camera occlusion is the most challenging problem.

In response to the above challenges of ambient light noise and the lack of access to the source code of high-precision Mocap systems, we propose a simple, yet open-source motion capture system named Easy Rocap, which uses markers of special material as the tracking objects of the mobile robot and uses multiple fixed cameras to perform 3D intersection to accurately estimate the position and orientation of drones and other robots in real time. Diverging from Mocap systems that rely on infrared cameras, our system takes indoor potential noise and obstacles into consideration, employing object filtering and multi-object tracking (MOT) algorithms that fuse object detection technology to ensure trajectories remain continuous and precise. Our main contributions are as follows.

(1) A Multi-view Correspondence method that combines Object Detection and MOT is proposed. The dual-layer detector achieves robust object detection in complex environments, and the MOT method can stably track markers under short-term

occlusion. Multi-view correspondences use geometric constraints to correctly match corresponding image points.

(2) A high-precision 3D robotic pose estimation system for complex dynamic scenes is open-sourced. When there is no obstruction in the simulation environment, the average positioning accuracy reaches 0.008 m, the average orientation accuracy reaches 0.65 degrees, and the solving speed reaches 30 frames per second (FPS).

The motion capture system proposed in this paper offers real-time pose information for robots in the experimental area. To assess its performance, extensive experiments were conducted using multiple cameras, including simulated environments and real scenes. The structure of the paper is outlined as follows. Section 2 presents the related works. Section 3 describes the system comprehensively, detailing keypoint detection, multi-view correspondence, and multi-view camera triangulation, respectively. Section 4 presents the experimental settings, results, and discussion. Section 5 provides a summary of the approach and offers a prospect for future work.

## 2. Related Works

This section provides an in-depth analysis of different motion capture techniques, comparing their distinct features. Additionally, algorithms for visual MOT and relative positioning are categorized and reviewed.

### 2.1. Motion Capture Methods

Motion capture (Mocap) serves as a digital method to track and record the spatial movements of targets [23], often employed in capturing and reconstructing the human body posture. The essential of Mocap is acquiring the 3D coordinates of each key point [24,25]. Mocap technology is widely applied in human activities [26,27] such as sports and entertainment, as well as robotic exploration fields for trajectory tracking and controlling [28].

In recent years, several types of Mocap systems have emerged. Depth cameras, for instance, can calculate depth by measuring the time delay between emitted light and the detection of its backscatter [29]. There are other systems such as inertial Mocap [30], mechanical Mocap, acoustic Mocap, and electromagnetic Mocap systems [24,25], but they are not as widespread as traditional Mocap systems, mainly due to deficiencies in accuracy or convenience.

Predominantly, traditional mocap technologies rely on optical approaches. Optical Mocap systems use high-speed cameras to capture reflective markers on the target object and triangulate the positions of these markers to reconstruct the object's spatial posture with high precision. In the field of motion capture, optical Mocap systems are often referred to as the benchmark for accuracy [24,25] and perform better in robotics applications. In scenarios such as human–robot interaction and local multi-robot collaboration [31,32], visual Mocap systems can address the common issue of drifts in inertial sensors. However, camera-based Mocap systems face challenges such as occlusions in certain camera views [23]. To overcome these problems, some researchers have designed hybrid systems that combine different Mocap technologies [33,34] to improve accuracy and reduce camera occlusions. As previously mentioned, autonomous control of drones requires high-frequency position and orientation information, so the structure should be as simple and efficient as possible.

Currently, marker-based optical motion capture systems are pretty mature in the market, with Vicon being a representative Mocap system [35]. However, it is difficult to apply and popularize it in daily use since its hardware cost is high and the technology is not open source. Recently, the open-source release of the EasyMocap toolkit has made low-cost human motion capture possible. At the same time, advancements in the precision and efficiency of deep learning algorithms have demonstrated their ability to improve the capabilities of Mocap systems, particularly in areas like object detection [36].

### 2.2. Multi-Object Tracking

Multiple Object Tracking (MOT) aims to identify and track multiple targets simultaneously over a period of time in processing sequential data [37]. Recently, a large number of MOT approaches have emerged, among which deep learning approaches are increasingly prevalent [38].

Predominantly, the methods employed in MOT comply with a principle known as the "tracking by detection" paradigm [36,39,40]. These approaches typically execute object detection on every single frame of a video sequence and then convert the task into object association between adjacent frames. Simple Online and Real-time Tracking (SORT) [39] utilized the Kalman filter [41] and Hungarian Algorithm [42] for object association. Because SORT disregards the appearance characteristics of detected objects, it performs well only when the certainty of object state estimation is high. Wojke et al. [36] introduced cosine similarity as appearance similarity for tracking association. As a result, this method can track through longer periods of occlusion but struggles with challenges like motion blur. MOT approaches based on joint detection and embedding [43–45] allow sharing detection and embedding information in the model, and cross-scale association is feasible. Previous studies [46,47] represent pioneering efforts in applying Transformer to MOT, achieving notable improvements in tracking accuracy. However, they encounter significant challenges in addressing computational bottlenecks during deployment. Meanwhile, Chu et al. [48] introduced a cascade association framework to handle challenging tracking scenarios, thereby improving calculation efficiency. Zeng et al. [49] developed Transformer-based frameworks for MOT, enabling end-to-end tracking, but its performance in detecting newborn objects is limited.

It is appropriate to select the corresponding approach according to the specific task requirements since these MOT approaches have different advantages. In our work, we have access solely to the preceding frame and the frame currently in view since pose estimation requires a real-time online tracker. Therefore, we use the mature and representative detection-based MOT approach OC-SORT [50], a tracking methodology that demonstrated top-tier effectiveness when tested on the KITTI dataset [51].

### 2.3. Direct Pose Estimation

We classify pose estimation into direct and indirect methods. Direct pose estimation directly observes the target to determine its pose, relying solely on raw sensor data without external infrastructure (e.g., GPS, UWB with anchors) or environmental feature matching. Although direct pose estimation techniques typically require customized hardware, their autonomous operation, reliability, and precision continue to draw substantial interest. Cutler et al. [52] proposed a straightforward yet efficient method by designing positioning markers composed of infrared LEDs, aiming to estimate the position and velocity of known markers. Faessler et al. [53] proposed an infrared-LED-based pose estimation system that utilizes the Perspective-n-Point (PnP) algorithm to compute the relative position and orientation between the UAV and UGV, enhancing the collaborative working efficiency between aerial and ground robots. However, to maintain the distinctness of the LED spots, these methods often only work at a short distance.

Ultra-Wideband (UWB) technology presents a promising solution. Fishberg and How [54] developed a 3-DOF relative 2D pose estimation system among robots, achieving improvements in mean position error solely based on UWB measurements. Cossette et al. [55] introduced a method for estimating relative orientations using UWB distance measurements, demonstrating that estimation accuracy can be increased by optimizing the formation geometry of robot formation. To reduce the impact of robot formation geometry and UWB measurement errors on pose estimation, Jones et al. [56] modified the state vector using a kinematic bicycle model, thereby facilitating the direct estimation of the lead vehicle's longitudinal speed and effective steering angle. Jin and Jiang [57] proposed a multi-vehicle mapping system that combines LiDAR, IMU, and UWB technologies to improve the robustness of positioning and mapping in degraded environments through

multi-metric weights LiDAR-inertial odometry and pose estimation correction from the degeneration direction. Hao et al. [58] proposed the KD-EKF algorithm, which enhanced the accuracy and consistency of cooperative localization in multi-robot systems by addressing the observability issues in the standard EKF approach.

In addition to integrating UWB technology, Pritzl et al. [59] introduced a cooperative guidance method for UAVs in GNSS-denied environments, leveraging the fusion of LiDAR and Visual-Inertial Odometry (VIO) data for precise relative localization and trajectory tracking. In a diverse team with a major UAV carrying LiDAR and a minor UAV equipped with a camera, the method combines LiDAR relative positioning data with the VIO output on the major UAV to accurately determine the minor UAV's pose.

## 3. Methods

To demonstrate our Easy RoCap system, the overall motion capture workflow is shown in Figure 1. The system is built using consumer-grade commercial IR RealSense (https://www.intelrealsense.com/depth-camera-d455/ (accessed on 17 June 2020)) cameras and consists of three major components: two-stage key points detection model (TKDM), Multi-view correspondences, and Multi-view Camera Triangulation.

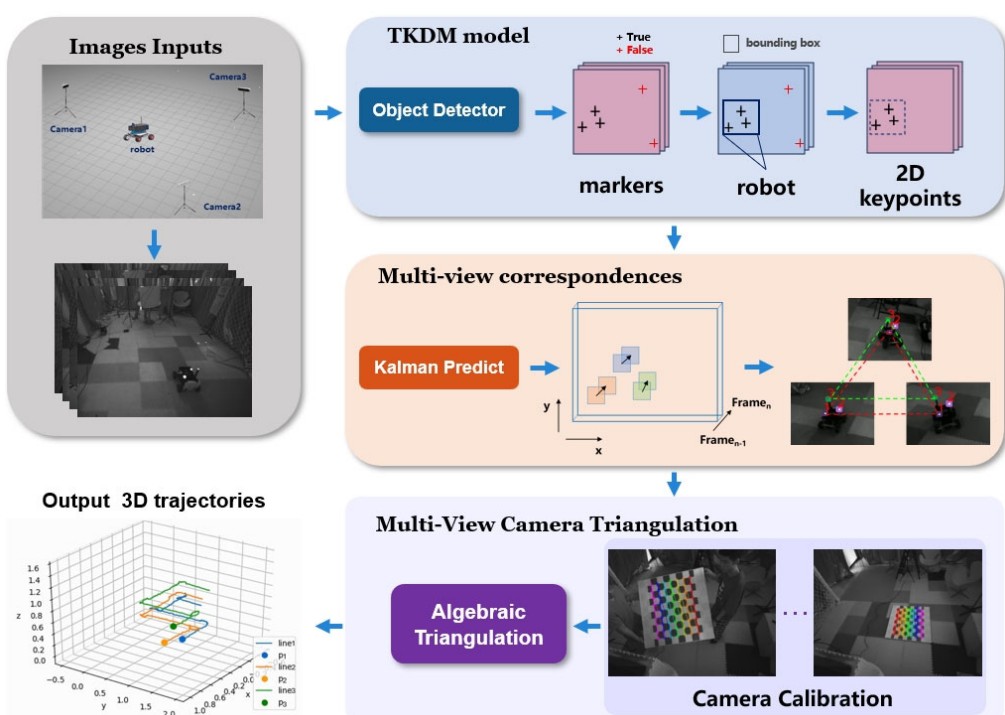

**Figure 1.** The workflow of Easy Rocap.

Firstly, a trained object detector is employed to generate bounding boxes of robots and markers in each view. Considering that there is noise in 2D detection, a filtering algorithm based on confidence and class is proposed to eliminate inaccurate detections. The multi-view correspondences module employs an MOT algorithm to address occlusion issues and assigns matched marker IDs. Utilizing the calibrated multi-view camera parameters, the 3D coordinates of markers are calculated, and the position and orientation of the robot are ultimately estimated by the coordinates of multiple markers.

### 3.1. Two-Stage Keypoints Detection Model (TKDM)

Accurate single-camera-based marker detection is essential for achieving precise multi-view localization. To minimize the ambient light interference while meeting real-time pose requirements for robots, we employ the YOLOX detector [60] known for its rapid detection speed to implement efficient positioning of markers. The decoupled head

strategy [61] effectively resolves the common conflict between classification and regression tasks by dedicating separate branches within the detector architecture to each specific task. By utilizing the decoupled head strategy, the detection speed of YOLOX is significantly enhanced without compromising accuracy. Furthermore, the multi-scale feature fusion technique in YOLOX is able to effectively improve detection performance in the presence of small objects and minor occlusions. Trained on a specific dataset, the detector produces detection boxes of different categories on image data.

Addressing the demand for simultaneously rapid inference across data from multiple views, the batch size is adjusted to satisfy that the quantity of images for inference aligns with the number of cameras, thereby accelerating image inference speed while aligning timestamps across multiple views.

---

**Algorithm 1** Filtering of Object Detection Results

---

   **Input**: frame $f_k$; object detector $Det$; detection score threshold $\tau$
   **Output**: Filtered marker detection boxes
1. $D_{remain} \leftarrow \varnothing$
2. $D_k \leftarrow Det(f_k)$
3. marker class boxes $D_m$, robot class boxes $D_r$
4. **for** $d$ in $D_m$ **do**
5.     Check if $d.score > \tau$ and $d$ is contained by a certain a robot box
6.     **if** Failed **then**
7.         delete $d$
8.     **end**
9.     $D_{remain} \leftarrow D_{remain} \cup \{d\}$
10. return: $D_{remain}$

---

Although it is already the most advanced detector in performance, there are still redundant or false detections in the real-world scene. To reduce the interference, detection results are filtered based on a strategy of taking the detection boxes of the robot as a constraint and filtering false detections. Considering that the markers are fixed on the robot target, the specific implementation is to use confidence to filter the object detection results and then take the robot detection boxes as a class label constraint to filter out false detections caused by environmental noise, enhancing the robustness of the detector. The specific algorithm is described in Algorithm 1, which is used to filter out invalid marker detection boxes.

$$det_{filtered}.score = \left\{ \begin{array}{l} det.score, \text{ if } det.score > \text{thre \& isincluded}(det) \\ 0, \quad \text{else} \end{array} \right. \tag{1}$$

where $det$ represents the marker detection box; $det.score$ is the score of the corresponding detection box; $det_{filtered}.score$ represents the score of the filtered detection box. For each marker detection box $det$, the confidence is first checked to see if it exceeds the threshold value *thre*. The function isincluded($det$) indicates whether the detection box is contained within the robot detection box, which is used to filter out false detections.

### 3.2. Multi-View Correspondences

Before computing the 3D pose of a robot, 2D object bounding boxes across multiple views should be appropriately matched, i.e., the bounding boxes of the same marker across all views are needed. Due to occlusion and motion blur, there might be missing detection, which is challenging. To address this, a tracking algorithm assigns IDs to objects and saves status information, preserving the trajectory continuously. Matching of IDs across multiple views is then facilitated based on this.

The task of MOT is to estimate the trajectories of multiple objects across successive video frames. In real-time online MOT, only the detection results from the latest few frames are available, making tracking by detection the leading paradigm in MOT. An increasing number of MOT methods are utilizing more powerful detectors to achieve better tracking

performance, with the YOLO series being widely applied due to its balance of speed and accuracy [40]. This paper builds upon the YOLO series detectors, with the tracking algorithm improved based on OC-SORT. Its characteristics include a fast tracking speed and robust handling of occlusions and non-linear motion [50]. Due to overlapping targets and non-linear motion, the Kalman filter fails to re-match with the current detection results, resulting in frequent track interruptions. OC-SORT proposes Observation-Centric Recovery (OCR) to recover the track based on the detection results rather than incorrect estimations. As shown in Figure 2, when a target with a lost track is redetected, the track is restored by associating the observations at the time of loss with the current observations. Experimental testing in this paper demonstrates that it is simple and online. In addition, to prevent errors in subsequent 3D calculations caused by incorrect detections, we establish additional conditions for tracker trajectory establishment.

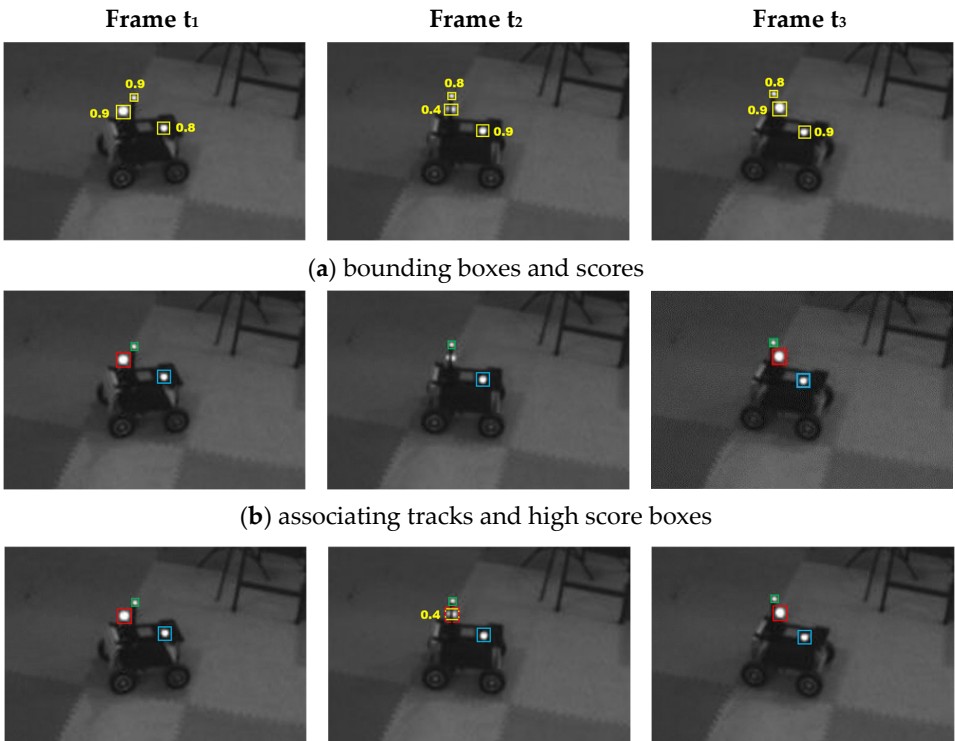

**Figure 2.** Instance of our MOT method based on observation. (**a**) The original detection results including confidence scores. (**b**) The trajectories of the algorithm before the improvement, which is only associated with the high-confidence detection boxes. The identical box color signifies the specific individual. (**c**) The utilization of low-confidence bounding boxes during object occlusion, and the red dashed box indicates the predicted position of the occluded target. Low-confidence detection boxes are successfully matched to previous unmatched trajectories.

As shown in Algorithm 2, for the matching of object detections and trajectories, the number of trajectories is constrained to be consistent with the number of IDs and markers. For example, if $n$ target markers are used, the IDs can only be 1, 2, $n$..., Additionally, when there are gaps in trajectory IDs, a new trajectory is established for targets that are successfully detected in three successive frames. Markers that are not updated temporarily do not participate in subsequent trajectory calculations until the next update.

---

**Algorithm 2** Trajectory Establishment

---

    **Input**: Track $T$; Filtered detection boxes $D_{remain}$
1. Associate $T$ and $D_{remain}$ using Similarity
2. $D_{re-remain} \leftarrow$ remaining object boxes from $D_{remain}$
3. $T_{remain} \leftarrow$ remaining tracks from $T$
4. **for** *det* in $D_{re-remain}$ **do**
5.     Check if there is an available $ID_{traj} \leq num_{market}$
6.     **if** True **then**
7.         $T_{remain} \leftarrow T_{remain} \cup \{det\}$
8.     **end**
9.     **else**
10.        delete *det*
11.     **end**
12. return: $T_{remain}$

---

An important cue to associate multiple objects across multiple views is that their associated joints should satisfy geometric constraints. Suppose $p \in \mathbf{R}^{N \times 2}$ denotes the 2D distribution of N object points, the geometric constraints between view $p_i$ and another view $p_j$ can be quantified using the following function.

$$E_g\left(p_i, p_j\right) = \frac{1}{2N} \sum_{n=1}^{N} d_g\left(p_i^n, L_{ij}\left(p_j^n\right)\right) + d_g\left(p_j^n, L_{ji}(p_i^n)\right) \tag{2}$$

where $p_i^n$ denotes the position of the *n*-th marker in view *i* and $L_{ij}\left(p_j^n\right)$ denotes the epipolar line (a fundamental concept within epipolar geometry, see [61] for definition) associated with $p_j^n$ from the other view. $d_g(a, l)$ signifies the distance from the point *a* to the straight line *l*. The scale of the error function $E_g$ is used to measure the geometric similarity scores corresponding to multiple views.

$$A_{match} = \underset{A}{\arg\min} \sum_{0 \leq i < j \leq m} E_g\left(p_i, p_j\right) \tag{3}$$

Suppose there are *m* cameras in the scene; *A* records the matching relationship of multi-view marker IDs. $A_{match}$ is obtained by minimizing the sum of functions $E_g$, so as to calculate the 3D coordinates of different markers directly.

*3.3. Multi-View Camera Triangulation*

After the cameras are correctly placed in the 3D space, the Easy Rocap system is then calibrated. Based on the detection and tracking results mentioned previously, this section mainly describes how to calibrate multi-view cameras and perform triangulation on the objects.

To reconstruct 3D positions using multi-view cameras, it is essential to rectify distortions produced by the camera lens and establish a geometric model for multiple cameras to denote their relative positions. The primary goal of calibration is to obtain both the intrinsic and extrinsic parameters of the cameras. The methodology adopted in this experiment is based on Zhang's camera calibration method [62]. The camera calibration yields intrinsic and extrinsic parameters, allowing us to establish a camera model. This means we can find the relationship between the spatial points' 3D coordinates and the camera image coordinates.

$$AX = 0 \tag{4}$$

Here, *X* is the augmented 3D coordinates of a spatial point and *A* represents the matrix combination of image coordinates and camera parameters [63].

While conventional triangulation methods utilize linear algebra [63] to estimate 3D coordinates from 2D coordinates across multiple views, challenges arise when certain views do not provide reliable 2D estimations due to occlusions or out of frame. To address this

issue, this study draws inspiration from methods employed in multi-camera multi-person motion capture systems [64]. The 2D positions together with the confidences of markers obtained by multiple cameras are passed to the algebraic triangulation module, which solves the triangulation problem in the form of a weighted linear equation:

$$(w_j \circ A_j)X_j = 0 \tag{5}$$

where $w_j$ denotes the confidence vector of the $j$-th target marker across multiple views; $A_j$ is a matrix combination of the image coordinates and camera parameters for the $j$-th object point across different views; and $X_j$ is the 3D position of the $j$-th target marker.

By employing the Singular Value Decomposition (SVD) method to solve the system of triangulation linear equations, optimal 3D coordinates for the markers are obtained. The rigid body formed by the markers fixed on the robot aligns with the robot's position and orientation. Through the calculation of the centroid translation and rotation matrix of the 3D point coordinates, the robot's pose variations can be accurately represented.

## 4. Experiments

In this section, we showcase several experiments to illustrate the precision and rapidity of our Easy Rocap system. We designed a series of experiments using UAVs and UGVs and employed the evo toolbox to assess the pose accuracy. The experimental procedure is separated into two segments: (1) UAV trajectory accuracy verification in the simulation environment and (2) UGV and UAV trajectory accuracy verification in real scenarios.

### 4.1. Experimental Settings

#### 4.1.1. Dataset Description

Due to the limited availability of datasets for evaluating robot poses, the experimental data were collected from carefully crafted simulation scenarios as well as real-world situations. Figure 3 shows the experimental sites with optical markers arranged on the surface of the unmanned drone/vehicle. The simulation scene dataset includes fixed multi-camera data with timestamps and ground truth trajectory data. The real-scene dataset includes fixed camera data with timestamps, as well as LiDAR data, which are used for SLAM methods to obtain trajectories for comparison.

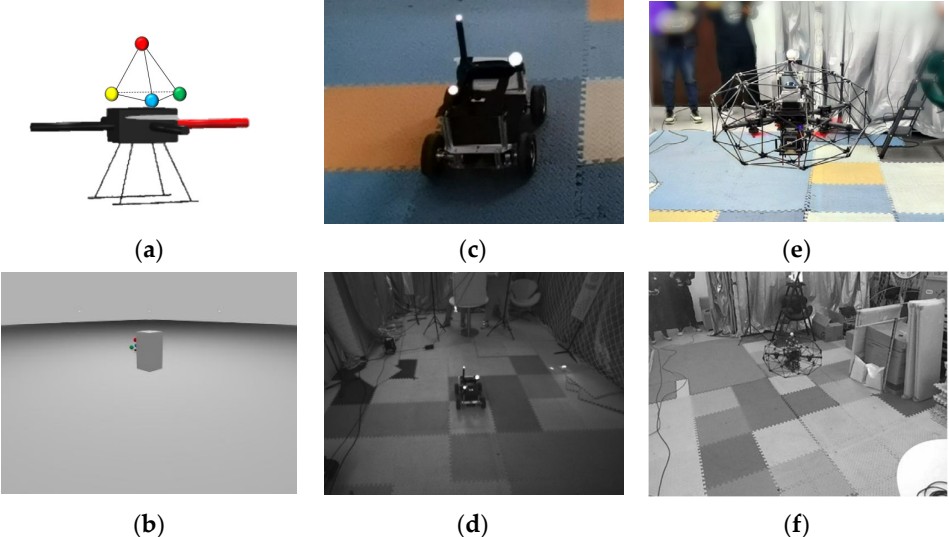

**Figure 3.** Test robot platforms and scenarios. (**a**) Simulated drone with markers. (**b**) Drone in simulation scenario (occlusion). (**c**) UGV with markers. (**d**) UGV in real scenarios. (**e**) Drone with markers. (**f**) Drone in real scenarios.

We collected a total of 1200 sample images in the simulation and real scenarios as the training dataset, including drones, UGVs, and markers. On the basis of recording the

calibration board data to obtain camera intrinsic and extrinsic parameters, we recorded data from different scenes and motion states, called Seq 01–06. Seq 01–03 data are recorded in the simulation scenario, where drones flew with and without obstacles, performing motions such as spirals. The average velocity is 0.4 m/s and the total length of the trajectory is 55.18 m. Seq04–05 data are recorded in the real scenario, where UGV performs curved and square movements. The average velocity is 0.26 m/s and the total length of the trajectory is 12.5 m. Seq06 data are recorded in the real scenario where the drone is performing curved movements. Table 1 shows the scene type, motion pattern, data length, data frequency, and average speed of the data.

**Table 1.** Data descriptions.

|  | Platform | Scene | Motion Pattern | Data Length/Seconds | Camera Frequency/FPS | Truth Traj Frequency /Hz | LiDAR Frequency /Hz |
|---|---|---|---|---|---|---|---|
| Seq01 | UAV | simulation | horizontal | 60.71 | 15 | 1000 | |
| Seq02 | UAV | simulation | helical | 44.81 | 15 | 1000 | |
| Seq03 | UAV | simulation | horizontal | 46.33 | 15 | 1000 | |
| Seq04 | UGV | real-world | curvilinear | 6.01 | 90 | | 10 |
| Seq05 | UGV | real-world | square | 42.80 | 90 | | 10 |
| Seq06 | UAV | real-world | curvilinear | 27.09 | 90 | | 10 |

### 4.1.2. Experimental Scenes

To validate the feasibility of our robotic motion capture system, a simulation experiment was first designed. As shown in Figure 4, the simulation experiment was conducted using the robot simulation software Gazebo, which provides high-fidelity physical simulation of the drone and the markers. A simulation experimental site of 10 m × 10 m was set up. Utilizing the Pinhole Camera Model, we customized camera parameters including the field of view and resolution. The cameras were positioned at a height of 3 m, as illustrated in Figure 4, with each camera tilted at an angle of 20°, facing towards the center of the venue. The drone was manually controlled for flight, and the software was able to record its ground truth trajectory.

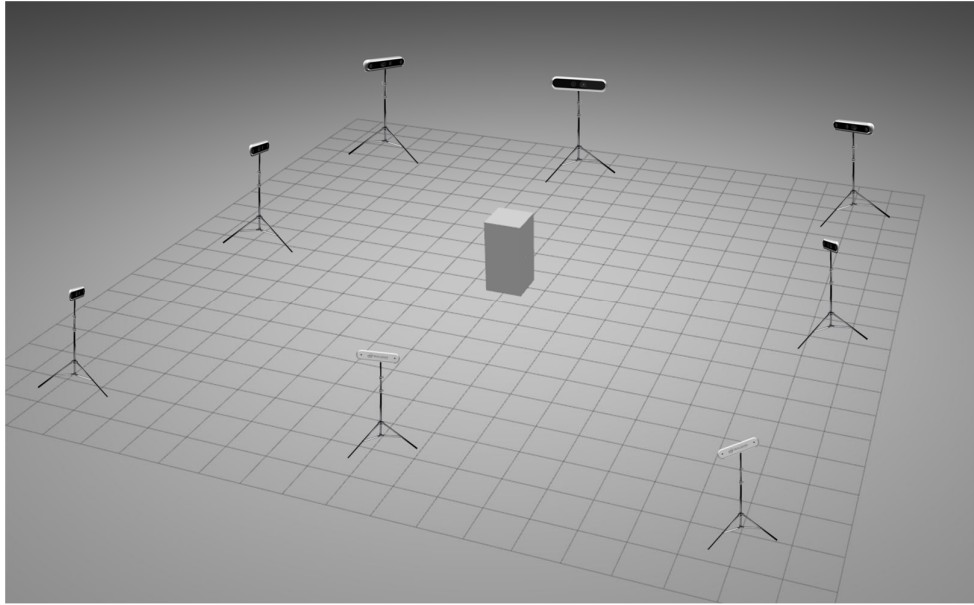

**Figure 4.** Simulation scenario.

In an effort to verify the stability of our Easy Rocap system, a respective experiment was organized in real scenarios (Figure 5) in comparison with the LiDAR-Inertial SLAM algorithm. The camera system consists of ix infrared lenses from Intel RealSense D455. The

experimental site consisted of UGV and UAV platforms with markers, both equipped with a small-sized Livox Mid-360 LiDAR sensor.

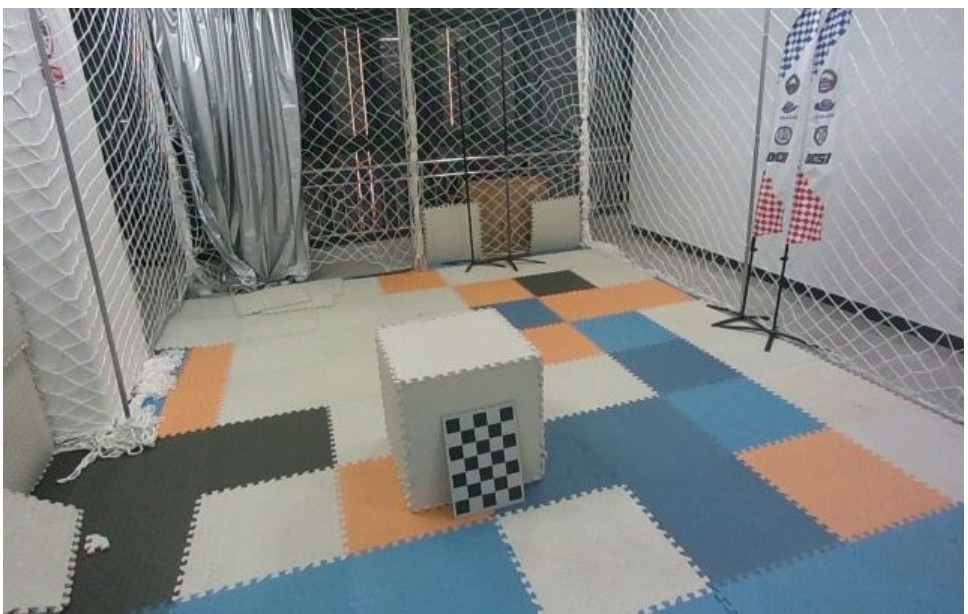

**Figure 5.** Real scenario.

For multiple object tracking of markers, the detector is YOLOX [60], and YOLOX-s was used as the backbone. The pretrained model was downloaded, and the categories were modified to "UGV", "UAV", and "marker". The training schedule is 1000, the number of training data is approximately 1000, and the input image size is 640 × 480. The process of data augmentation encompasses techniques such as Mosaic [65] and Mixup [66]. These strategies serve the purpose of improving the model's generalization ability. Regarding the technical setup of the model's training phase, this process was conducted on an NVIDIA GeForce RTX 2070 graphics processing unit (GPU), with a batch size of 4 and a total training time of approximately 4 h.

4.1.3. Evaluation Metrics

In assessing the precision of our Easy Rocap system, we use an error evaluation method (https://github.com/MichaelGrupp/evo (accessed on 14 September 2017)). The reference in the simulation scenario is the ground truth trajectory recorded in Gazebo, and the reference in the real scenario is the pose estimation result of Easy Rocap. For the evaluation of trajectory accuracy, the global consistency of the trajectories was assessed utilizing absolute pose error (APE) as a metric. As illustrated in Equation (6), the calculation of APE was made based on the estimated trajectory posture $P_{est,i} \in SE(3)$ at timestamp $i$, and the ground truth trajectory $P_{ref,i} \in SE(3)$ at timestamp $i$.

$$E_i = P_{est,i}\Theta P_{ref,i} = P_{ref,i}^{-1}P_{est,i} \in SE(3) \tag{6}$$

where the operator $\Theta$, deriving the relative pose when given two separate poses, signifies the inverse compositional process.

In order to show the accuracy of our system in translation and rotation, we used the corresponding two indicators, $APE_{trans}$ and $APE_{rot}$, respectively. The translation accuracy indicator and rotation accuracy indicator are calculated by Equations (7) and (8), respectively. On this basis, we used the APE of each timestamp, i.e., $APE_i$ to calculate the average $APE$, $APE_{mean}$, and the root mean square error (RMSE) $APE_{RMSE}$ of all timestamps

to measure the accuracy of the overall trajectory, as shown in Equations (9) and (10), where the total count of timestamps is represented by N.

$$APE_{trans,i} = \|trans(E_i)\| \tag{7}$$

$$APE_{rot,i} = \left| angle\left( \log_{SO(3)}(rot(E_i)) \right) \right| \tag{8}$$

$$APE_{mean} = \frac{1}{N}\sum_{i=1}^{N} APE_i \tag{9}$$

$$APE_{RMSE} = \sqrt{\frac{1}{N}\sum_{i=1}^{N} APE_i^2} \tag{10}$$

### 4.2. Experimental Results

In order to showcase the effectiveness of our approach, we undertook experimental tests on the dataset. First, we considered the calculation time: the detector's detection time for each frame is 0.02 ms, and the multi-view triangulation time for each frame is 0.01 ms.

Next, we executed simulation experiments to assess the precision of our technique. As shown in Figure 6 and Table 2, the outcomes of the experiment indicate our technique's ability to supply consistent and real-time position data with great precision. Data Seq01 and Data Seq02 are collected in the unobstructed environment. The average translational accuracy ranges from 0.69 to 0.79 cm, achieving millimeter-level precision. The trajectory orientation error is less than 0.65°, validating the correctness of our algorithm. Data Seq03 is collected during fast drone flight around the obstacle, with an average position estimation error of 1.32 cm.

**Table 2.** Results of simulation experiments.

| | Scene | Motion Pattern | Trajectory Length/m | Duration /s | RMSE(APE) /m | Mean (APE) /m |
|---|---|---|---|---|---|---|
| Seq01 | simulation | horizontal | 23.73 | 60.71 | 0.0080 | 0.0069 |
| Seq02 | simulation | helical | 12.79 | 43.91 | 0.0090 | 0.0079 |
| Seq03 | simulation | horizontal (occlusion) | 18.67 | 46.33 | 0.0150 | 0.0132 |

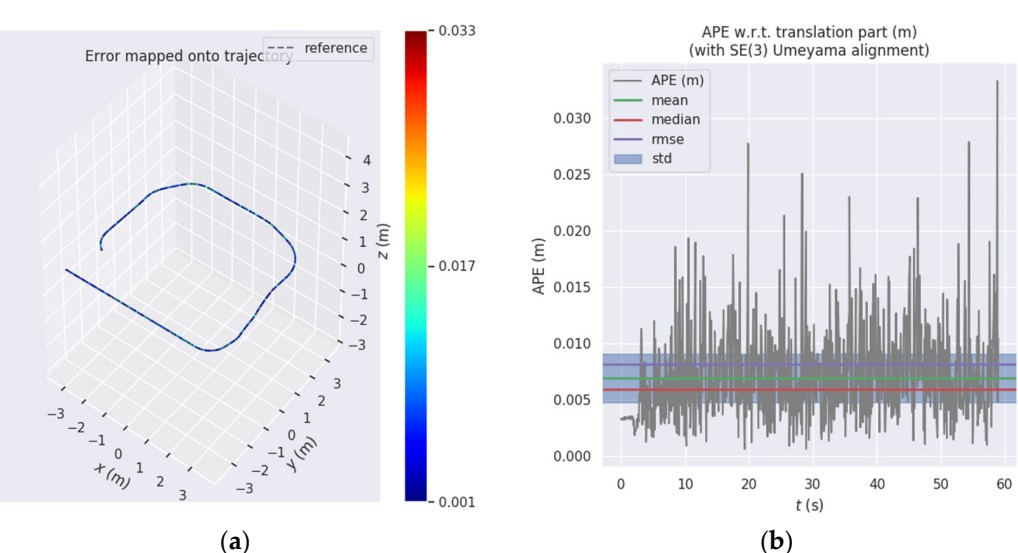

**(a)**        **(b)**

**Figure 6.** *Cont.*

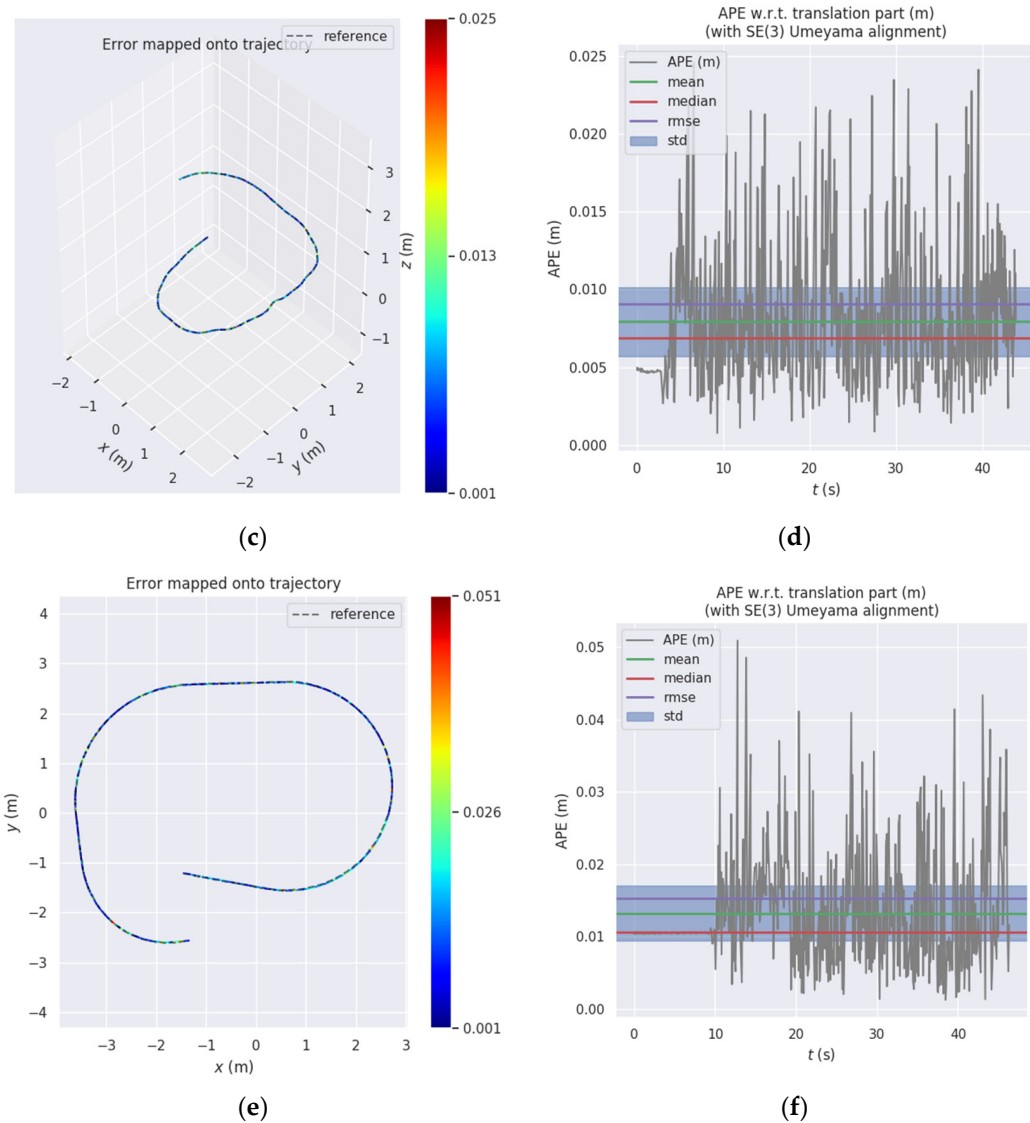

(c)

(d)

(e)

(f)

**Figure 6.** Results of Seq01−03 trajectory evaluation. (**a**) Trajectory Plot of Seq01. (**b**) APE Plot of Seq01. (**c**) Trajectory Plot of Seq02. (**d**) APE Plot of Seq02. (**e**) Trajectory Plot of Seq03. (**f**) APE Plot of Seq03.

The experimental results of Seq01–03 are compared in Figure 7. In an unobstructed environment, the system achieves millimeter-level accuracy in 3D pose estimation. However, the precision of the trajectory is impacted by any occlusions on specific markers. It is evident that the displacement of detection bounding boxes for the object can influence the accuracy of the 3D pose estimation. This is also the reason why the overall trajectory APE of Seq03 is larger than that of Seq01–02.

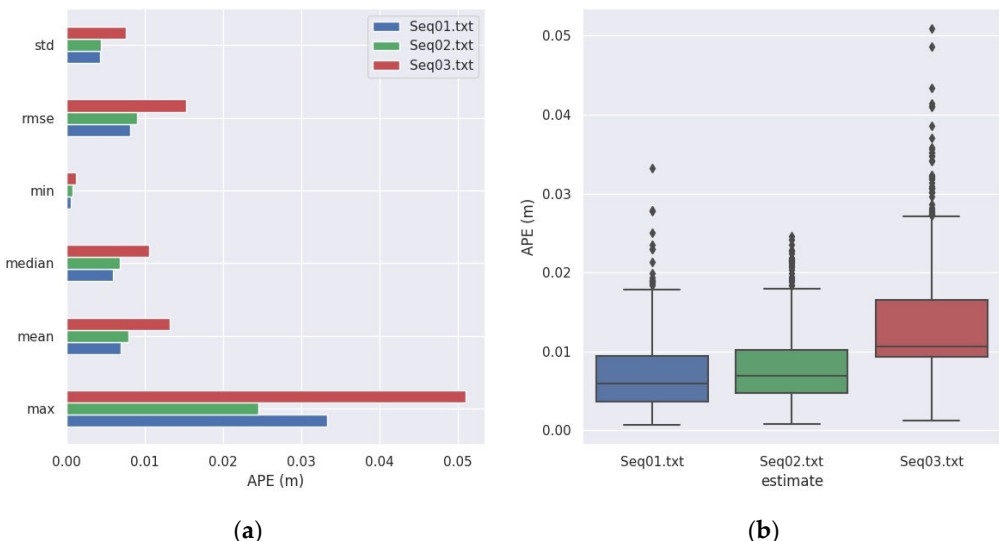

**Figure 7.** Statistics results for APE on Seq01−03. (**a**) Comparison histogram of APE for Seq01−03. (**b**) Box plot of APE for Seq01−03.

In order to show the effectiveness of our system in real-world scenarios, we carried out an experiment comparing it with LiDAR-Inertial SLAM. As shown in Figure 8a,b, using Data Seq04–06, the pose estimation results obtained from Easy Rocap were compared with those from FAST LIO2 [67]. In the test on Seq04, when the trajectory length of curved motion is short, the average trajectory error estimated by the two methods is 4.6 cm. When compared to the accurate distance measured by a physical ruler, from the starting to the endpoint, the difference in distance noted by our system is 4.7 cm, as opposed to LiDAR-Inertial SLAM's significantly larger difference of 14.1 cm. This also verifies the fact that LiDAR-Inertial SLAM systems may experience drifts during indoor turning. As shown in Figure 8c,d, Data Seq05 is a square trajectory for a UGV. By setting the result of Easy Rocap as the reference trajectory, it is clear that our method generates a trajectory that is smoother and more continuous in turns, while the SLAM method is more likely to experience drift when making sharp turns and accumulate significant errors. Figure 8e presents the actual height (Z-axis) of the trajectories of the ground vehicle, derived from LiDAR-based SLAM and our Easy Rocap, respectively. To explain, the discrepancy in starting positions is due to the UGV moving a certain distance before the initiation of SLAM. The SLAM method exhibits significant jumping in the *z*-axis direction, which is not in alignment with the real scenario. In contrast, our system's calculated trajectory remains stable in the *z*-axis direction. Figure 8f,g show the flight trajectories of the drone from side and top views in the real-world scenario. The trajectory obtained by our system remains smooth and continuous, while the trajectory obtained by LiDAR-Inertial SLAM is more prone to drift when making sharp turns.

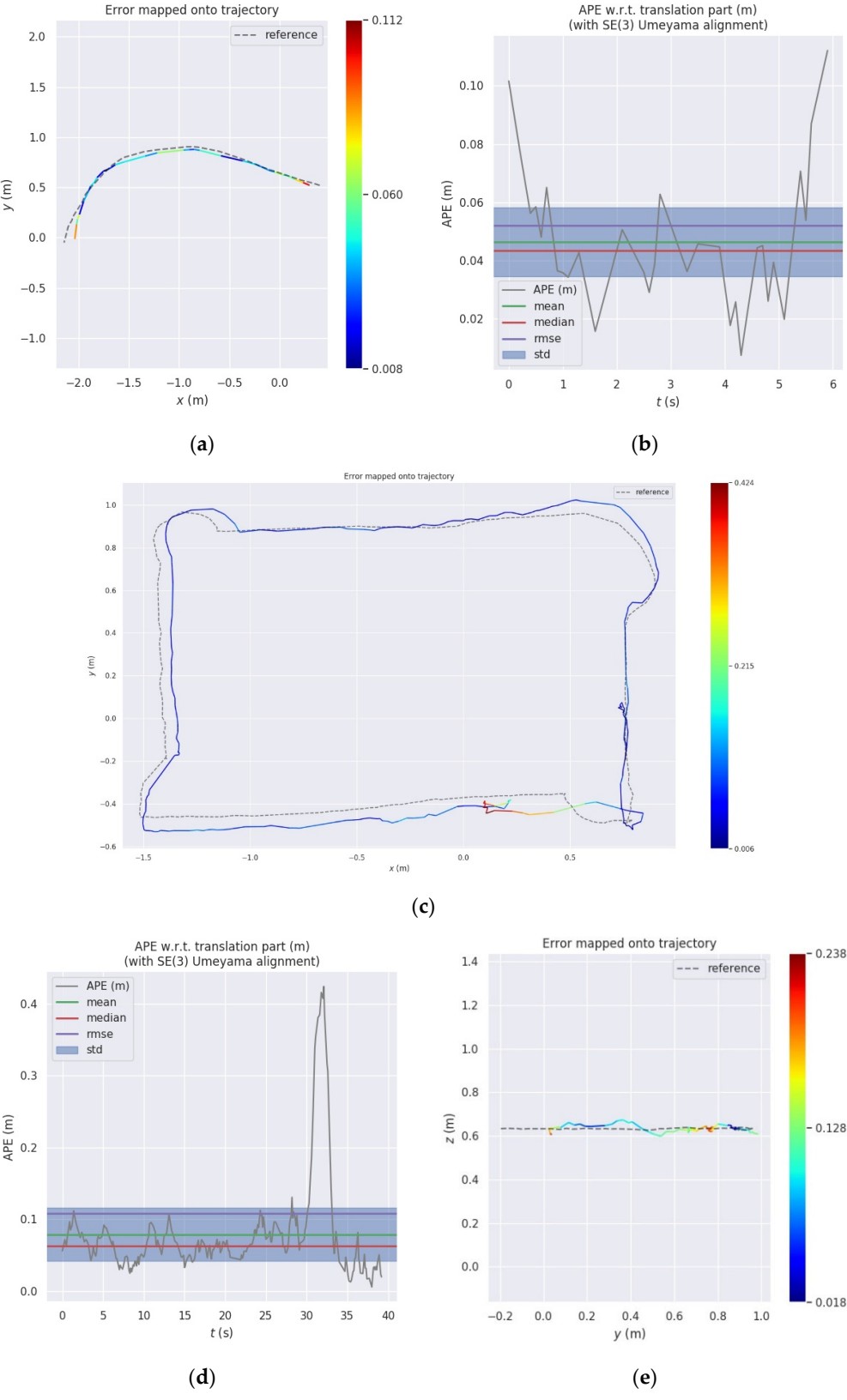

**Figure 8.** *Cont.*

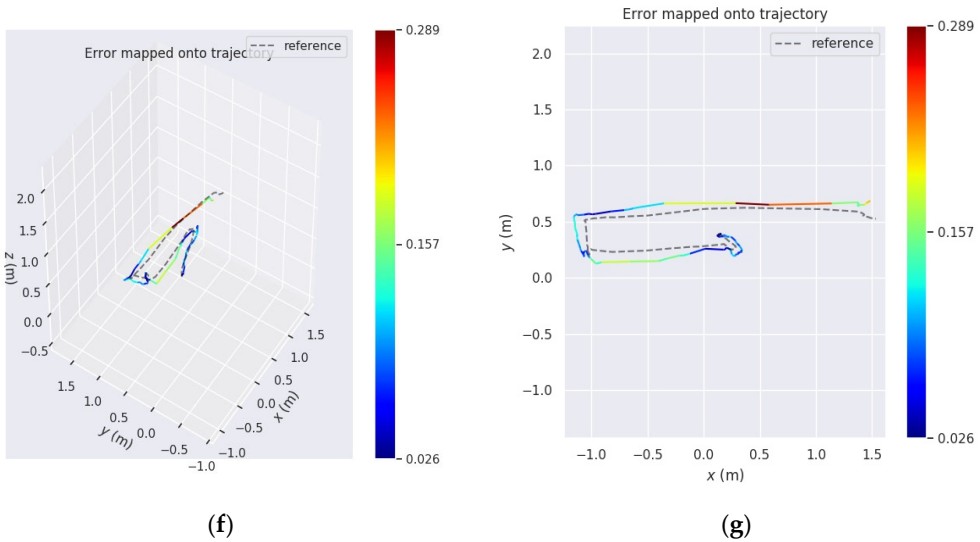

(**f**)                                                              (**g**)

**Figure 8.** Results of Seq04–06 trajectory evaluation. (**a**) Trajectory Plot of Seq04. (**b**) APE Plot of Seq04. (**c**) Trajectory Plot of Seq05. (**d**) APE Plot of Seq05. (**e**) Comparison Plot of Z-Axis Trajectories for Seq05. (**f**) Trajectory Plot of Seq06. (**g**) APE Plot of Seq06.

*4.3. Discussion*

This paper presents a cost-effective and easy-to-use robotic pose capture system. Our hardware system consists solely of a computer equipped with a graphics processing unit (GPU), three or more Intel RealSense D455 cameras, and several infrared marker balls. Compared to commercial high-precision optical motion capture systems on the market, which can cost upwards of tens or even hundreds of thousands of dollars, this setup significantly reduces cost. The system's ease of use is mainly reflected in its robustness, eliminating the need for users to have a strong professional background.

A broad and varied dataset used in the experiments showcases the precision and flexibility of the proposed method. Firstly, the system's accuracy and speed were effectively demonstrated in the experiments conducted using simulated scenarios. The average translational error between the trajectories of this method and the ground truth is less than 0.8 cm, and the calculation speed can reach 30FPS. Experiments in the simulated scenario have also verified that our method can accurately calculate poses even in the presence of non-linear motion or short-term occlusions. The strong performance of the methodology is closely linked to the combination of techniques like object detection and MOT. Secondly, compared to the real distance between the start and end points, our method achieves higher localization accuracy than SLAM. In the end, our system's robustness was demonstrated in comparative experiments conducted in real-world scenarios. Compared to advanced SLAM methods, the trajectory obtained by the method is smoother, more continuous, and less prone to drifting. Additionally, there is no accumulation of errors in our method.

However, in order to improve the speed, the chosen MOT algorithm relies on a kinematic model for frame-to-frame correlation, and it does not record the appearance information of the objects. Therefore, our method may not be able to handle long-term occlusions. Nevertheless, we have implemented a response mechanism for long-term occlusions, where multi-view correspondences are repeated when the lost target reappears. This response mechanism may take less than 0.5 s. This mechanism improves the robustness of the robot's pose estimation. In addition, there is a trade-off between camera resolution and transmission speed, and camera resolution also affects the effective area of the Rocap system. One of the limitations associated with cameras is the potential distortion at the lens edge, which can slightly affect the accuracy of the 3D coordinates when the robot moves to the edge of most cameras' field of view. In the future, the research will combine high-performance computers with high-resolution cameras to further expand the effective area of the Rocap system.

## 5. Conclusions

This study proposes an affordable and easy-to-use robotic motion capture system, which can precisely capture the location and orientation of an unmanned platform in complex scenes. The project's code will be open-sourced on Github (https://github.com/DCSI2022/EasyRocap (access after 1 June 2024)). To address the interference of environmental light noise on visual detection, a TKDM detection model with fine filtering is proposed. Unlike traditional tracking by detection paradigm MOT algorithms that may use wrong estimates when associating tracks, the MOT method used in this paper relies more on observations. To reduce the effects of occlusion from a certain view on the 3D pose estimation results, weights are added to the matrix coefficients of different views based on confidence, ensuring that the contributions of each view are reasonable. The system, composed of commercial-grade cameras, was tested under various situations and motion types, demonstrating high precision and fast speed. The average translation accuracy for pose estimation in the simulation environment is less than 0.8 cm, and the orientation accuracy is less than 0.65 degrees, with a solving speed of 30 FPS. In real scenarios, compared with advanced SLAM algorithms, the practical applicability is verified. The results show that our system can overcome the drift and accumulated errors of the SLAM method. The integrated system only requires data streams from calibrated cameras to estimate robot poses, making it convenient to deploy in robot systems. In addition, there are potential enhancements in future work. In one respect, we will focus on improving the accuracy of our system while maintaining speed by utilizing high-performance computers and high-resolution cameras. In another respect, we will explore marker-less methods for estimating robot 3D poses using 2D feature maps.

**Author Contributions:** Conceptualization, H.W. and C.C.; methodology, H.W. and C.C.; software, H.W.; validation, H.W.; formal analysis, H.W. and Y.H.; investigation, H.W. and Y.H.; resources, H.W., C.C., S.S. and L.L.; data curation, H.W., S.S. and L.L.; writing—original draft preparation, H.W.; writing—review and editing, H.W., Y.H. and Y.X.; visualization, H.W. and Y.H.; supervision, C.C.; project administration, C.C.; funding acquisition, C.C. and B.Y. All authors have read and agreed to the published version of the manuscript.

**Funding:** This research was funded by the National Natural Science Foundation of China (No. U22A20568), the National Key RESEARCH and Development Program (No. 2022YFB3904101), the National Natural Science Foundation of China (No. 42071451), the Natural Science Foundation of Hubei, China (No. 2022CFB007), the Key RESEARCH and Development Program of Hubei Province (No. 2023BAB146), the National Natural Science Foundation of China (No. 42130105), the National Key RESEARCH and Development Program-Key Special Projects for International Cooperation in Science and Technology Innovation between Governments (No. 2019YFE0123300), the Fundamental Research Funds for the Central Universities, and the European Union's Horizon 2020 Research and Innovation Program (No. 871149).

**Data Availability Statement:** The experimental data and the corresponding code of the Easy Rocap system will be available at https://github.com/DCSI2022/EasyRocap (access after 1 June 2024).

**Acknowledgments:** The authors would like to thank the Supercomputing Center of Wuhan University for providing the supercomputing system for the numerical calculations in this paper.

**Conflicts of Interest:** The authors declare no conflicts of interest.

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
