# Peer review of "Easy Rocap: A Low-Cost and Easy-to-Use Motion Capture System for Drones"

_drones, doi:10.3390/drones8040137_

Round 1
Reviewer 1 Report
Comments and Suggestions for Authors
See attached.

Comments on the Quality of English LanguageAuthor Response
Please see the attachment.

Reviewer 2 Report
Comments and Suggestions for Authors
The paper is devoted to indoor robot motion capture system, working for both UAVs and UGVs. Solution is relatively low-cost, open-source and designed to work in real time.
This paper can be of certain interest for scholars in the field of object detection, object tracking, applied geometry and trigonometry and those working with robotics.
The problem under consideration is relevant at the present time, due to the increased interest in drones, UAVs etc. Having a solution which is low-cost, real-time and open-source is always appreciated.
Paper’s abstract describes the relevant science field and its challenges. As well as gives all the relevant information about the system being developed, technologies being used and results. Introduction gives a more detailed look on technologies, methods and challenges relevant to the paper. Second chapter compares different motion capture techniques and algorithms, citing all the relevant sources.Third chapter describes the system developed and the methods used. Forth chapter describes both simulated and real experiments and results obtained. All of the evaluation metrics are described, and formulas are present if/where necessary.
Conclusion addresses the significance of the developed system and evaluates the results obtained.
Algorithms, figures, tables and formulas are understandable, sufficient and relevant.
However, I have several questions for the Authors. Please, find them below.
1. Please, consider removing GitHub link from the abstract (line 31), since you have it in (line 490)
2. «IR Mocap» in (line 82) means infrared. Please specify this in brackets or add “(IR Mocap)” to the line 79, when you first mention this, like you did it in (line 70) with Mocap.
3. Section 2.2 (lines 148-170) contains a great variety of MOT algorithms. Please, consider adding advantages and disadvantages for some of them, because only naming different approaches feels unnecessary.
4. Please, provide additional information and link for KITTI dataset in (line 169).
5. «VIO» appears in (line 197) and never explained. It may be unintuitive for those who are not working with robotics.
6. Please, describe «decoupled head strategy», when it first brought up in (line 221).
7. Specify which algorithm you are referring to in (line 276), instead of using «pseudocode» term
8. Mention Algorithm 2 where it is relevant, because for now it is not mentioned anywhere.
9. Authors used different number of cameras for real and simulated scenarios. How this (different number of cameras) affects the results?
Minor issues:
1. Consider removing word «clearly» in (line 203).
2. Figure 1 is too wide and some elements are to small (axes in «3d animate» and indexes of frames in «Multi-view correspondences»). Please, consider reallocating them in horizontal 2 lines instead of one.
3. Please, separate explanation from the name of Figure 1 in (line 213).
4. Please remove the ending "-es" in line 268 for «we establishes», therefore changing it to «we establish» or «we established».
5. Consider fitting Table 1 and 2 width to the text width.
Nevertheless, I believe the reviewed manuscript can be a great contribution to MDPI Symmetry after relatively minor revisions.
Thank you for the possibility to read and review this interesting manuscript.
Round 2
Reviewer 1 Report
Comments and Suggestions for Authors
I find the paper acceptable in its current form.
Comments on the Quality of English LanguageNone
Author Response
Dear Reviewer:
Thank you for the positive comments on the manuscript. We greatly appreciate your time and effort in reviewing our work. We have carefully reviewed the manuscript and have made revisions to enhance the clarity and readability. These revisions include grammatical corrections, adjustments in sentence structure, and clarification of certain phrases to ensure that our intended meaning is accurately conveyed.
All revisions are highlighted in red in the manuscript. Thank you once again for your valuable feedback and support.
Best wishes!